# Scaling Robot Policy Evaluation via Discrete Diffusion World Model

**Yaxuan Li** [1]   **Junjie Wen** [2]   **Zhongyi Zhou** [1 2]   **Yefei Chen** [1]   **Chaomin Shen** [1]   **Yaxin Peng** [3]   **Yichen Zhu** [2]

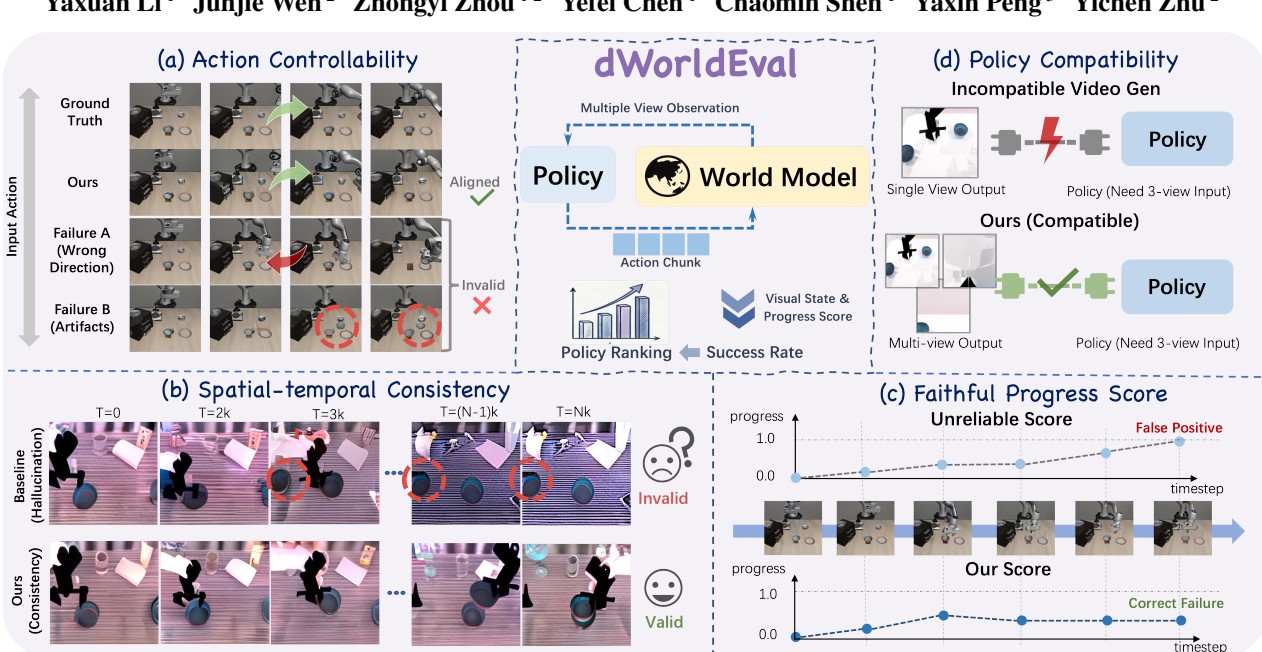

*Figure 1.* **dWorldEval** is a discrete diffusion world model designed to be controllable and consistent for reliable policy evaluation. It ensures action controllability and spatiotemporal consistency while providing the policy compatibility and faithful progress scoring necessary for accurate, automated policy ranking.

## Abstract

Evaluating generalist robot manipulation policies is costly and difficult to scale in the real world. While emerging world models (e.g., WorldE-val, Ctrl-World) offer a promising alternative, the reliability of such evaluation remains a critical bottleneck. Specifically, their visual predictions can undermine policy assessment by "self-correcting" failures into false positives or yielding artifacts under out-of-distribution controls. Even with failure-enriched data, current architectures struggle to capture action-causal dynamics, as they typically treat actions as passive conditions

rather than causal drivers. To address this, we propose dWorldEval, an action-centric discrete-diffusion world model that maps visual observations, language instructions, and action chunks into a shared unified token space and denoises them with a single self-attention backbone where actions function as first-class tokens. To realize reliable policy-world interaction, dWorldE-val introduces a sparse keyframe memory that anchors global scene state while preserving fine-grained multi-view interaction cues, and leverages Progress-as-text to jointly generate future obser-vations and success indicators. Extensive exper-iments on LIBERO, RoboTwin, and real-robot tasks demonstrate that dWorldEval significantly outperforms video diffusion baselines in action controllability, stabilizes long-horizon multi-view rollouts, enabling accurate policy ranking via au-tomatic success estimation.

[1]School of Computer Science,East China Normal Univer-sity, Shanghai, China [2]Current Robotics, Shanghai, China [3]Department of Mathematics & School of Future Technol-ogy, Shanghai University, Shanghai, China. Correspondence to: Chaomin Shen <cmshen@cs.ecnu.edu.cn>, Yichen Zhu <yichen_zhu@foxmail.com>.

*Proceedings of the $43^{rd}$ International Conference on Machine Learning*, Seoul, South Korea. PMLR 306, 2026. Copyright 2026 by the author(s).

# 1. Introduction

While generalist robot manipulation policies have advanced rapidly (Black et al., 2024; Brohan et al., 2023; Kim et al., 2024; Hu et al., 2023; Liu et al., 2025; Physical Intelligence et al., 2025; Kim et al., 2025; Gemini Robotics Team et al., 2025a; Bjorck et al., 2025; Zhao et al., 2025a;b; Zhen et al., 2024), evaluating their capabilities remains a significant challenge. Consequently, generative world models have emerged as a scalable alternative to costly real-world execution or asset-heavy simulations for evaluating robot policies (Li et al., 2025b; Guo et al., 2025b; Quevedo et al., 2025; Gemini Robotics Team et al., 2025b; 1X World Model Team, 2025; Huang et al., 2025; Jiang et al., 2025; Tseng et al., 2025).

However, world models have not yet become reliable evaluation proxies for robotics policies, primarily because they often fail to accurately reflect robot actions and physical interactions. We attribute these failures to two main causes. First, existing models struggle to generalize to out-of-distribution (OOD) actions. Since they are typically trained on successful demonstrations, they tend to ignore erroneous actions and hallucinate successful outcomes due to a distribution shift. Second, physical inconsistency leads to unrealistic artifacts. For instance, rigid objects may visually warp during contact or vanish entirely due to spatiotemporal inconsistencies.

While existing works attempt to mitigate this by incorporating failure trajectories, the effectiveness is limited as action coverage is infeasible (Guo et al., 2025b; 1X World Model Team, 2025). We argue that the bottleneck is fundamentally architectural. Most existing approaches adapt architectures originally designed for video generation (e.g., image-to-video models). Since these backbones are not natively designed to take robotic actions as input, actions are merely injected as auxiliary conditions (e.g., via cross-attention or adaptive modulation like AdaLN) into the visual denoiser (Li et al., 2025b; Guo et al., 2025b; Quevedo et al., 2025). Given that these backbones are heavily pre-trained on massive video datasets, they inherit strong visual priors. Consequently, action signals act as weak guidance and are frequently overridden by these dominant priors, leading to hallucinated success or spatiotemporal drift.

Motivated by this, we propose dWorldEval, a world model based on Masked Discrete Diffusion (MDD) (Austin et al., 2021; Sahoo et al., 2024; Lou et al., 2023; Nie et al., 2025; Li et al., 2025a; Yang et al., 2025). Unlike pre-trained video backbones, dWorldEval is trained from scratch on robotic data, treating actions and visual observations as equivalent tokens to ensure action controllability. Specifically, We map visual observations, language instructions, and action chunks into a unified token space, modeling them jointly via a self-attention backbone. To enable reliable policy evaluation, we incorporate a sparse keyframe memory that maintains spatiotemporal consistency by mitigating long-horizon drift. Additionally, we introduce a discrete progress token to quantify task completion; by jointly predicting this token with future observations, the model automatically determines success when progress reaches 1.

In summary, we make three contributions:

- We propose dWorldEval, a discrete-diffusion world model that significantly enhances action controllability, utilizing sparse keyframe memory to ensure spatiotemporal consistency.

- We jointly predict visual outcomes and a discrete progress token to enable automatic success detection.

- We conduct a systematic evaluation on LIBERO (Liu et al., 2023), RoboTwin (Mu et al., 2025), and real-world tasks. Extensive experiments confirm that dWorldEval achieves substantially better action controllability measured by our proposed action-sensitive $\Delta$-LPIPS metric. Furthermore, its estimated success rates correlate strongly with actual execution performance (Pearson $r \approx 0.9$), enabling accurate ranking of policies across capabilities.

# 2. Related Work

**World models for policy evaluation.** While policy evaluation has traditionally depended on real-world rollouts (Li et al., 2024; Zhou et al., 2025) or physics-based simulators (Mees et al., 2022; Todorov et al., 2012; Mu et al., 2021; Gu et al., 2023; Xiang et al., 2020; Makoviychuk et al., 2021; Mittal et al., 2023; Yu et al., 2020; Mu et al., 2025; Sferrazza et al., 2024; Grotz et al., 2024; Liu et al., 2023), world models offer a data-driven paradigm to scale policy assessment (Li et al., 2025b; Guo et al., 2025b; Quevedo et al., 2025; Gemini Robotics Team et al., 2025b; 1X World Model Team, 2025; Huang et al., 2025; Jiang et al., 2025; Tseng et al., 2025). However, the utility of current video-based evaluators is severely limited by their lack of reliability. Fundamentally, these architectures treat actions as auxiliary conditions into a visually-dominated denoiser—e.g., via AdaLN modulation in WorldGym (Quevedo et al., 2025) or cross-attention in Ctrl-World (Guo et al., 2025b) and WorldEval (Li et al., 2025b). This design allows strong visual priors to override control signals, causing the model to frequently hallucinate transitions. In contrast, dWorldEval integrates actions as primary tokens within a unified sequence, enabling the generated future states to be directly driven by control signals via self-attention.

**Discrete diffusion in robotics.** Discrete diffusion language models (DLMs) have emerged as strong alternatives

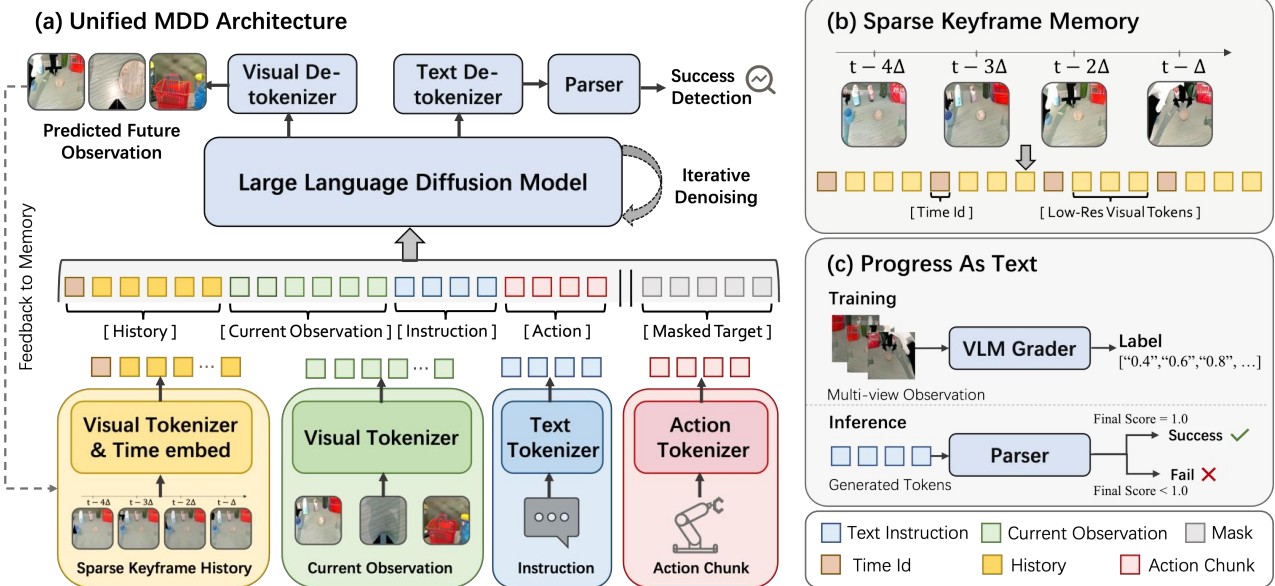

*Figure 2.* **Overview of dWorldEval. (a) Unified Architecture:** Diverse modalities are flattened into a single sequence, enabling the model to treat visual, control, and semantic tokens uniformly. **(b) Sparse Keyframe Memory:** A history of low-resolution keyframes is used to anchor global spatiotemporal consistency. **(c) Progress-as-text:** The model jointly predicts visual outcomes and discrete progress tokens to enable automatic success detection.

to autoregressive LLMs, exhibiting competitive generative capabilities while enabling flexible sampling strategies and improved controllability (Austin et al., 2021; Sahoo et al., 2024; Lou et al., 2023; Nie et al., 2025). Recent efforts extend DLM backbones to multimodal understanding and generation, e.g., LaViDa (Li et al., 2025a) and MMaDA (Yang et al., 2025). In robotics, dVLA (Wen et al., 2025a) and related works adapt discrete diffusion to policy learning, formulating action prediction as token inpainting over VLA-style inputs (Wen et al., 2025a; Liang et al., 2025; Wen et al., 2025c). In contrast to these policy learning approaches, we employ discrete diffusion to build a world model. Our unified token space enables joint prediction of visual observations and progress scores, eliminating the need for external VLMs or reward functions required by prior evaluators (Li et al., 2025b; Quevedo et al., 2025; Guo et al., 2025b).

# 3. Methodology

## 3.1. Problem Formulation

**World model formulation.** We aim to learn a world model $\mathcal{W}_\theta$ that serves as a proxy for evaluating robotic manipulation policies. Specifically, given a global language instruction $l$, a current observation $o_t$, a history $h_t$, and a sequence of future actions $\mathbf{a}_t$, the model predicts the visual outcome $\hat{o}_{t+\Delta}$ and a task progress score $\hat{v}_{t+\Delta} \in [0, 1]$ at horizon $\Delta$. Essentially, $\mathcal{W}_\theta$ approximates the joint distribution of future visual dynamics and task completion: $(\hat{o}_{t+\Delta}, \hat{v}_{t+\Delta}) \sim \mathcal{W}_\theta(\cdot \mid o_t, \mathbf{a}_t, h_t, l)$.

**Prerequisites for reliability.** To serve as a reliable evaluator, $\mathcal{W}_\theta$ must satisfy three properties: (1) Action Controllability: predictions must faithfully reflect the visual changes induced by the input actions; (2) Spatiotemporal Consistency: the model must preserve the spatial layout and consistency across long-horizon rollouts via the history context $h_t$; and (3) Discriminative Task Completion: generated future states must be semantically unambiguous to allow for accurate success detection.

**Policy evaluation via imagination.** We define an imagined rollout as a closed-loop interaction between a policy $\pi$ and the world model $\mathcal{W}_\theta$, yielding a generated trajectory $\tau = \{\hat{o}_0, \hat{o}_\Delta, \ldots, \hat{o}_T\}$. The policy's performance is evaluated by the imagined success rate $J = \frac{1}{N} \sum_{i=1}^{N} \mathcal{S}(\tau_i)$, where $\mathcal{S}(\cdot) \in \{0, 1\}$ is a success indicator function that judges whether the generated outcome fulfills the task instruction.

## 3.2. World Modeling via Discrete Diffusion

### 3.2.1. UNIFIED TOKEN SPACE FOR DIRECT ACTION CONDITIONING

We employ specialized tokenizers to map each modality into a shared discrete latent space: MAGVIT-v2 (Yu et al., 2023) for RGB observations, the LLaDA tokenizer (Nie et al., 2025) for language instructions, and FAST (Pertsch et al., 2025) for continuous action chunks $\mathbf{a}_t$. To unify these diverse representations, tokens from different modalities

are mapped into a single index space via fixed offsets and serialized into a flattened sequence. Crucially, rather than processing actions as separate conditions, this joint sequence passes through a unified backbone with bidirectional attention. This design enables visual generation targets to attend directly to action tokens at every layer. We posit that such dense interaction treats actions as tokens of equal standing with visual observations, facilitating finer-grained modeling of action-causal dynamics and ensuring strict action controllability.

### 3.2.2. SPARSE KEYFRAME MEMORY FOR SPATIOTEMPORAL CONSISTENCY

To satisfy the spatiotemporal consistency requirement (Sec. 3.1) while balancing context length and computational efficiency, we employ a multi-resolution design conditioned on a sparse keyframe memory. History keyframes are encoded solely from the primary global view (e.g., the top-down view) at a reduced resolution, whereas the current observation retains high-resolution multi-view details. Drawing inspiration from recent multimodal architectures like Seed-1.5VL (Guo et al., 2025a). We further prepend explicit time-index tokens to each keyframe to provide absolute temporal positioning. Functionally, this asymmetry separates responsibilities: the low-resolution history constrains global scene geometry and object permanence, while the high-resolution current context supplies the interaction cues necessary for precise action-causal generation.

### 3.2.3. PROGRESS-AS-TEXT FOR AUTOMATIC SUCCESS DETECTION

We formulate task progress estimation as discrete text generation within the unified token space, avoiding separate regression heads. This co-generation of visual outcomes and progress values unifies generative and discriminative capabilities, grounding synthesis in task semantics. During training, we distill supervision from the SEED-1.5VL (Guo et al., 2025a) model. By prompting the VLM to grade visual transitions against task milestones (see Appendix C for prompts and examples), we obtain scores which are then converted into discrete numeric tokens (e.g., "1.0") and appended to the target sequence. At inference, the generated text tokens are parsed back into a scalar $\hat{v}_{t+\Delta}$, providing an intrinsic metric for autonomous policy ranking without external oracles.

### 3.3. Joint Visual-and-Progress Denoising

We employ conditional Masked Discrete Diffusion (MDD) to learn $p(\mathbf{y}_{t+\Delta} \mid \mathbf{c}_t)$, where only the target suffix is masked for reconstruction while the context $\mathbf{c}_t$ remains unmasked.

**Training objective.** Given context $\mathbf{c}_t$ and target $\mathbf{y}_{t+\Delta}$, we sample a diffusion level $\lambda \sim \mathcal{U}(0, 1)$, and corrupt the target by a mask-based forward process $\tilde{\mathbf{y}}_{t+\Delta} \sim q_\lambda(\tilde{\mathbf{y}} \mid \mathbf{y}_{t+\Delta})$. Let $\Omega_\lambda = \{j \mid \tilde{y}_j = \texttt{[MASK]}\}$ be the masked index set. We train the model to reconstruct the clean tokens at masked positions:

$$\mathcal{L}_{\text{WM}} = \mathbb{E}_{\tau, \lambda, \tilde{\mathbf{y}}} \left[ -\frac{1}{m(\lambda)} \sum_{j \in \Omega_\lambda} w_j \log p_\theta(y_j \mid \mathbf{c}_t, \tilde{\mathbf{y}}_{t+\Delta}, \lambda) \right],$$
(1)

where $m(\lambda)$ denotes the masking probability at level $\lambda$. and $w_j$ is a modality-specific rebalancing weight.

**Inference.** During evaluation, we employ iterative parallel decoding to sample the target $\mathbf{y}_{t+\Delta}$ conditioned on the unified context $\mathbf{c}_t$. This process simultaneously recovers the future visual state and its predicted score.

## 4. Experiments

In this section, we evaluate the generative capabilities of dWorldEval and its function as a robot policy evaluator. Specifically, our investigation addresses the following research questions:

- **(RQ1)** Does dWorldEval strictly adhere to control signals, faithfully rendering execution failures from suboptimal or OOD actions rather than hallucinating outcomes?

- **(RQ2)** Does the sparse keyframe memory (Sec. 3.2.2) effectively prevent spatiotemporal drift, thereby maintaining long-horizon consistency required for evaluation?

- **(RQ3)** Can the proposed Progress-as-text mechanism serve as an accurate intrinsic success metric at inference time without external oracles?

- **(RQ4)** Is dWorldEval a reliable proxy for assessing diverse robot policies? Specifically, does the performance estimated via closed-loop rollouts strongly correlate with real execution and accurately rank policies across different architectures and training stages?

- **(RQ5)** Are action controllability and spatiotemporal consistency strict prerequisites for policy evaluation? Does the $\Delta$-LPIPS metric effectively diagnose the loss of controllability that leads to unreliable evaluation?

### 4.1. Experimental Setup

**Platforms and data.** We conduct evaluations across three diverse platforms, configuring the training data for each to support world modeling:

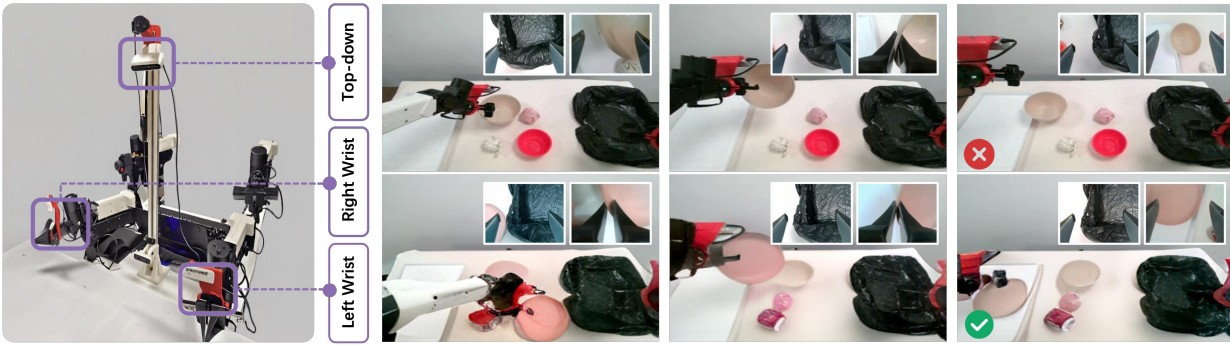

**Robot Setup**                                                                 **Generated Cases: Clean the table.**

*Figure 3.* **Real-world setup and rollout visualization. Left:** The bimanual AgileX platform equipped with three synchronized cameras. **Right:** Result of a failure and a success rollout generated for the *Clean the Table* task; the insets in the top corners of each frame display the synchronized wrist views.

**(1) LIBERO** (Liu et al., 2023): We utilize the LIBERO-Object, LIBERO-Spatial, LIBERO-Goal, and LIBERO-100 suites, featuring synchronized third-person and wrist views. To enable failure-aware scoring, we augment the 5.5k official expert demonstrations with 1k failed rollouts collected from suboptimal policies.

**(2) RoboTwin** (Mu et al., 2025): We select the ARX arm configuration to evaluate contact-rich tableware manipulation. The resulting dataset comprises 5.5k trajectories across 10 distinct tasks, ranging from multi-object stacking (e.g., blocks and bowls) to precise pick-and-place operations.

**(3) Real-world setup**: We deploy a physical bimanual AgileX system, as shown in Figure 3, featuring two 6-DoF arms, and recorded via three synchronized RealSense 457 cameras. Data collection is performed via teleoperation at 50 Hz. The dataset totals 5.2k trajectories (including 1k human-collected failures) across five tasks: Bussing Table, Place Cup, Handover Block, Strike Block and Place Bottles.

**Baselines and target policies.** We benchmark dWorldEval against video diffusion baselines WorldEval (Li et al., 2025b) and WorldGym (Quevedo et al., 2025), ensuring identical training data splits for fair comparison. And we assess its capabilities across varied policies: multiple training checkpoints of a base policy ($\pi_0$ (Black et al., 2024)) on LIBERO, and heterogeneous architectures (e.g., DexVLA (Wen et al., 2025b), Diffusion Policy (Chi et al., 2025)) across RoboTwin and real-world environments.

**Implementation details.** All models predict multi-view outcomes at $256^2$ resolution, conditioned on a sparse history of $K = 4$ keyframes ($128^2$). Regarding the loss weights in Eq. 1, we set $w_j = 2$ for progress tokens and $w_j = 1$ for visual tokens. The prediction horizon $\Delta$ aligns with the action chunk length (selected from $[2, 8]$). During inference, we employ 16-step iterative parallel decoding. We report success rates averaged over 20 episodes per task for

simulation benchmarks and 30 episodes for real-world experiments. Evaluating a full trajectory takes approximately 30–90 seconds (1.5s/frame) on a single H800 GPU.

**4.2. Evaluating the World Model**

4.2.1. EVALUATING ACTION CONTROLLABILITY

**Evaluation protocol.** To assess action controllability (RQ1), we employ a protocol on a test set constructed from two distinct interaction types: (1) Expert Success ($\mathcal{D}_{\text{succ}}$): successful trajectories collected from a fully trained policy; (2) Suboptimal Failure ($\mathcal{D}_{\text{fail}}$): failure rollouts collected from undertrained checkpoints. We perform generation conditioned strictly on ground-truth action sequences and evaluate on the shared third-person view.

**Evaluation metric.** Standard LPIPS captures only static appearance. To explicitly evaluate action controllability, we introduce $\Delta$-**LPIPS**, which measures the perceptual fidelity of state transitions rather than absolute states. For a fixed stride $\Delta$, we compute difference images $\Delta o_t = o_{t+\Delta} - o_t$ (analogously for predictions $\Delta \hat{o}_t$) and define:

$$\Delta \text{LPIPS} = \mathbb{E}_t \Big[ d_{\text{lpips}} \big( \text{norm}(\Delta \hat{o}_t), \text{norm}(\Delta o_t) \big) \Big], \quad (2)$$

where $\text{norm}(\cdot)$ denotes per-sample RMS normalization for stability. We use $\Delta$LPIPS as our primary indicator for action-conditioned dynamic fidelity (see Sec. 4.4 for further validation of its diagnostic correlation).

**Experimental results.** As shown in Fig. 4, dWorldEval accurately generates multi-view outcomes for both successful and failure executions. This visual fidelity is quantified in Tab. 1, which exposes a critical divergence: baselines suffer severe degradation on the Failure subset (e.g., WorldGym $\Delta$LPIPS spikes from 0.347 to 0.650), whereas dWorldEval maintains consistent performance. Such stability contrasts with the hallucinations observed in baselines (Fig. 5a). To

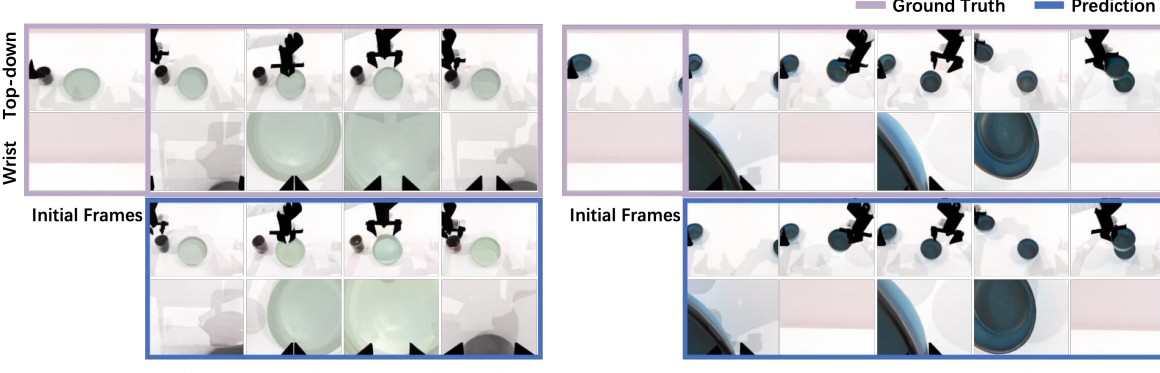

*Figure 4.* **Visualization of ground-truth vs. generated multi-view rollouts on RoboTwin** (Mu et al., 2025). **Left/Right**: suboptimal vs. successful execution. **Top/Bottom**: ground-truth simulator rollout vs. dWorldEval prediction, conditioned on identical action sequences. Each sequence displays the initial state followed by synchronized future frames.

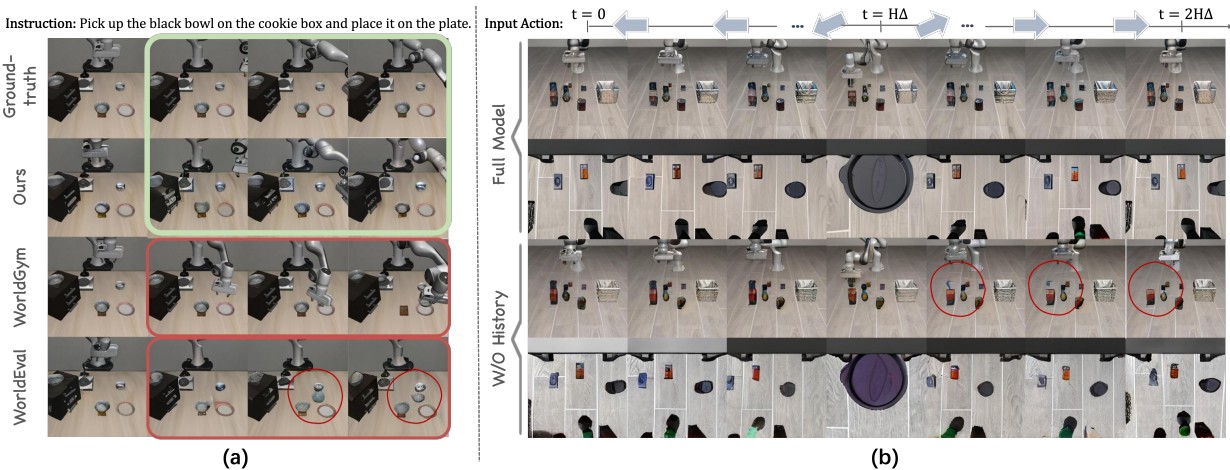

*Figure 5.* **(a) Action-controllability under suboptimal inputs.** We condition all methods on an unseen action sequence and compare their rollouts. While WorldGym (Quevedo et al., 2025) self-corrects the missed grasp into a successful pickup, WorldEval (Li et al., 2025b) hallucinates non-existent objects. In contrast, dWorldEval faithfully reproduces the failure. **(b) Long-horizon round-trip consistency.** We apply a reversible action trajectory: forward actions up to $t = H\Delta$, followed by the corresponding inverse actions to $t = 2H\Delta$. The full model returns close to the initial observation at $t = 0$, whereas removing history causes accumulated drift over the round trip.

*Table 1.* **Action controllability on LIBERO** (Liu et al., 2023). Comparison of LPIPS vs. our dynamic-aware $\Delta$LPIPS on Expert ($\mathcal{D}_{succ}$) and Failure ($\mathcal{D}_{fail}$) subsets. Baselines degrade on failure data, while ours remains consistent.

| METHOD | LPIPS ($\downarrow$) | $\Delta$LPIPS ($\downarrow$) | |
|---|---|---|---|
| | | EXPERT | FAILURE |
| WORLDEVAL | 0.262 | 0.423 | 0.701 |
| WORLDGYM | 0.218 | 0.347 | 0.650 |
| **OURS** | **0.215** | **0.315** | **0.352** |

*Table 2.* **Long-horizon consistency analysis.** We report the round-trip LPIPS ($\downarrow$) error, averaged across all synchronized views, over varying horizons $H$ to isolate the impact of memory. Without memory, the model suffers from progressive drift; with memory, spatiotemporal consistency is effectively preserved even at $H = 20$.

| METHOD | ACTION HORIZON ($H$) | | | |
|---|---|---|---|---|
| | 5 | 10 | 15 | 20 |
| OURS (W/O HISTORY) | 0.177 | 0.186 | 0.302 | 0.411 |
| **OURS (FULL)** | **0.130** | **0.145** | **0.193** | **0.243** |

### 4.2.2. EVALUATING SPATIOTEMPORAL CONSISTENCY

verify these results stem from causal control, we find that randomly shuffling actions (App. B) significantly degrades $\Delta$LPIPS, proving our model is sensitive to the input actions.

**Evaluation protocol and metrics.** To assess long-horizon stability (RQ2), we employ a variable-horizon round-trip protocol. We perform rollouts of length $2H$ by append-

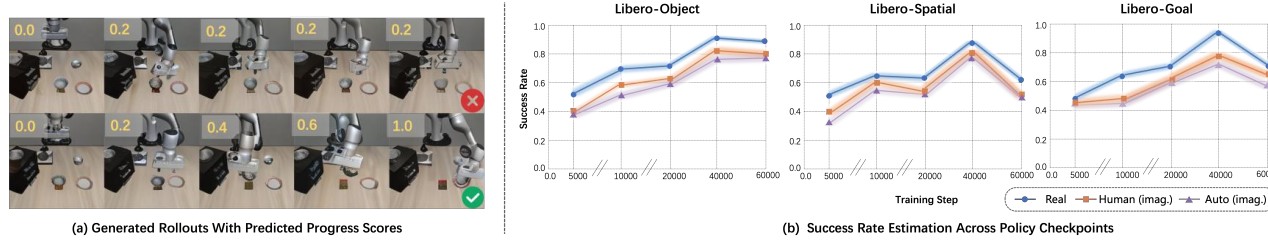

(a) Generated Rollouts With Predicted Progress Scores

(b) Success Rate Estimation Across Policy Checkpoints

*Figure 6.* **Joint generation enables automatic policy scoring.** **(a)** At each rollout step, the world model jointly predicts the future observation $\hat{o}_{t+\Delta}$ and a scalar progress score $\hat{v}_{t+\Delta} \in [0, 1]$, which finally indicates task success or failure. **(b)** Success rate estimates across checkpoints of a base policy $\pi_0$ (Black et al., 2024) on three LIBERO (Liu et al., 2023) suites. We compare ground-truth execution (*Real*) against evaluations on generated rollouts: human judgment based on the generated observations (*Human*) and automatic evaluation determined by the model's predicted progress score (*Auto*).

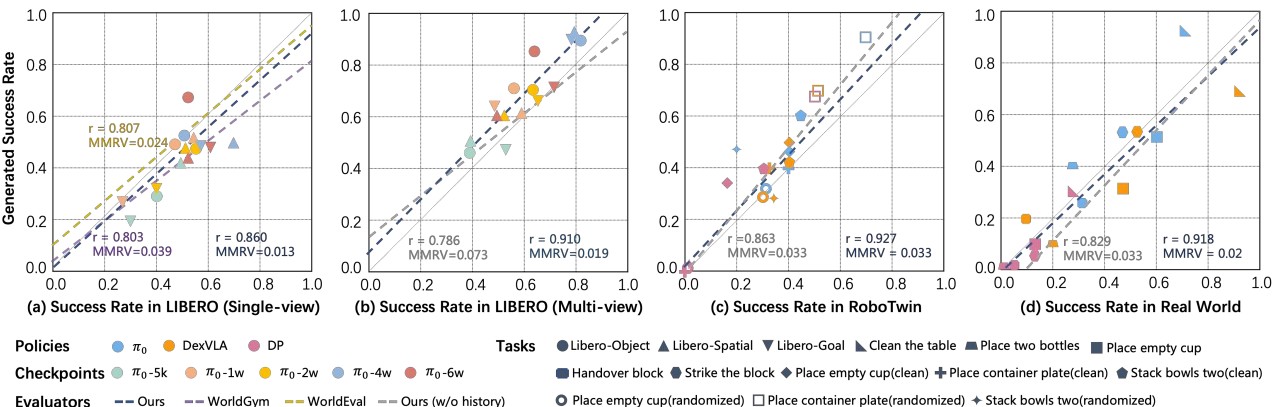

*Figure 7.* **Correlation between real-execution and world-model success rates.** Scatter plots compare real success rates (x-axis) against generated estimates (y-axis), reporting Pearson correlation $r$ and rank-violation MMRV (lower is better). **(a)** Comparison with video diffusion baselines (WorldEval (Li et al., 2025b), WorldGym (Quevedo et al., 2025)) on LIBERO (Liu et al., 2023) (Single-view). **(b-d)** Ablation studies comparing dWorldEval against its *w/o-history* variant across diverse settings: **(b)** LIBERO (Multi-view), **(c)** RoboTwin (Mu et al., 2025) with heterogeneous policies ($\pi_0$ (Black et al., 2024), DexVLA (Wen et al., 2025b), DP (Chi et al., 2025)), and **(d)** Real-world tasks. dWorldEval consistently achieves superior correlation and ranking accuracy across all domains.

ing inverse actions to trajectories of varying lengths $H \in \{5, 10, 15, 20\}$. Consistency is measured via LPIPS between the initial $o_t$ and final $\hat{o}_{t+2H}$. We focus on memory ablation here. Baselines are detailed in Appendix D, as their inability to strictly follow actions renders the round-trip metric unreliable for direct comparison.

**Experimental results.** Fig. 5(b) visualizes the structural drift isolated by removing memory. This is quantified in Tab. 2: while ablation errors accumulate as $H$ extends to 20, the full model maintains high fidelity, confirming that keyframe memory effectively bounds long-term drift. This stability is vital: Sec. 4.3 shows that drift leads to false negatives, severing the correlation with real-world performance.

### 4.2.3. JOINT SUBGOAL-AND-PROGRESS PREDICTION ENABLES AUTOMATIC VISUAL GRADING

**Evaluation protocol.** We leverage the failure-augmented training regime (Sec. 4.1) to validate whether the learned progress tokens can serve as an intrinsic metric (RQ3). We

evaluate multiple checkpoints of a base policy $\pi_0$ across three LIBERO suites and compare three success estimators: (1) **Real**: Ground-truth success rates measured by real execution; (2) **Human (imag.)**: Manual grading of the final generated frame; (3) **Auto (imag.)**: Our proposed automated metric, which classifies an imagined rollout as successful if the final predicted score reaches completion (i.e., $\hat{v}_{T+\Delta} = 1$).

**Experimental results.** As shown in Fig. 6(a), the progress score is discriminative, exhibiting sharp transitions upon task completion. Crucially, Fig. 6(b) demonstrates that Auto (imag.) closely tracks real execution, capturing even non-monotonic fluctuations (e.g., performance dips in later checkpoints). This alignment nearly matches human judgment and confirms that dWorldEval grounds its scoring in visual dynamics rather than monotonic priors, enabling accurate autonomous evaluation.

*Table 3.* **Universal multi-view fidelity.** Quantitative evaluation across diverse platforms (LIBERO (Liu et al., 2023), RoboTwin (Mu et al., 2025), Real-Robot) and synchronized viewpoints. dWorldEval maintains consistent high fidelity (low ΔLPIPS) on both simulation and real-world data.

| DATASET | VIEWPOINT | LPIPS (↓) | ΔLPIPS (↓) |
|---|---|---|---|
| LIBERO | 3RD-PERSON | 0.215 | 0.324 |
| | WRIST | 0.192 | 0.303 |
| ROBOTWIN | TOP-DOWN | 0.199 | 0.317 |
| | WRIST | 0.220 | 0.345 |
| REAL WORLD | TOP-DOWN | 0.272 | 0.365 |
| | WRIST | 0.254 | 0.336 |

*Table 4.* **Full consistency comparison.** We report the round-trip LPIPS (↓) error averaged over varying horizons $H$. Comparisons with WorldEval (Li et al., 2025b) and WorldGym (Quevedo et al., 2025) demonstrate that our model maintains superior spatiotemporal consistency, whereas baselines suffer from significant drift at longer horizons.

| MODEL | ACTION HORIZON ($H$) | | | |
|---|---|---|---|---|
| | 5 | 10 | 15 | 20 |
| WORLDEVAL | 0.216 | 0.320 | 0.467 | 0.531 |
| WORLDGYM | 0.191 | 0.224 | 0.308 | 0.482 |
| **OURS (FULL)** | **0.130** | **0.145** | **0.193** | **0.243** |

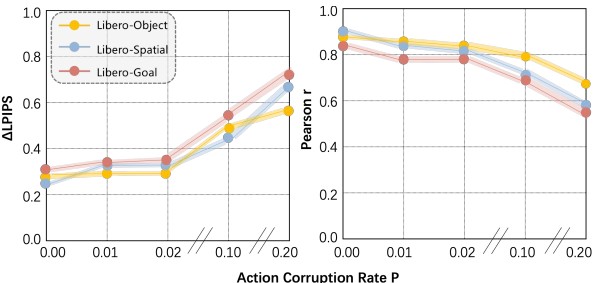

*Figure 8.* **Action corruption tests.** Ground-truth action chunks are replaced by chunks from other episodes with probability $p$. **Left:** ΔLPIPS increases with $p$, indicating degraded controllability. **Right:** Pearson correlation between real and estimated success rates drops accordingly.

## 4.3. World-Model as a Reliable Proxy for Policy Evaluation

To ensure fairness and prevent score-visual misalignment in drifting baselines when addressing RQ4, we determine success strictly based on generated terminal visual states rather than intrinsic scores.

**Comparison with baselines on LIBERO.** Fig. 7(a) evaluates policy ranking accuracy on LIBERO tasks under the single-view setting. Existing baselines (WorldGym and WorldEval) exhibit weaker correlations and higher rank violations (MMRV up to 0.039) due to insufficient action controllability. In contrast, dWorldEval achieves a strong linear correlation with minimal rank violation (MMRV=0.013), confirming that action-faithful generation is a prerequisite for reliable evaluation.

**Ranking heterogeneous policies across domains.** We further assess robustness across multi-view settings, heterogeneous architectures (RoboTwin), and physical environments (Real World). As shown in Fig. 7(b-d), the *w/o-history* ablation suffers from significant performance degradation, particularly in the multi-view setting ($r$ drops to 0.786). Conversely, dWorldEval effectively anchors spatiotemporal consistency, maintaining high correlations across LIBERO multi-view ($r = 0.910$), RoboTwin ($r = 0.927$), and real-world ($r = 0.918$) tasks, demonstrating superior generalization despite domain gaps and randomized settings.

## 4.4. More Experiments and Ablation Study

**Universal fidelity across diverse platforms.** We provide a comprehensive assessment of visual fidelity across diverse platforms. As detailed in Tab. 3, dWorldEval maintains consistently low ΔLPIPS scores ($\approx 0.31\text{-}0.36$) across varying camera configurations and robot morphologies. Notably, the performance in the real-world setup remains comparable to simulation results. This consistency suggests that our unified tokenization is robust against domain gaps, maintaining high fidelity across diverse environments.

**Is action-controllability strictly necessary?** To verify this prerequisite and validate Δ-LPIPS as an effective diagnostic metric (**RQ5**), We apply an inference-time intervention: swapping action chunks with probability $p$ to break alignment while preserving statistics. As shown in Fig. 8, as $p$ increases, ΔLPIPS worsens and the ranking correlation with real success rates degrades precipitously. This diagnostic proves that *only* in the low-ΔLPIPS regime (high controllability) can $\mathcal{W}_\theta$ accurately preserve policy ordering, confirming that faithful action modeling is the foundational mechanism for evaluation.

**Comparison of long-horizon consistency.** While Sec. 4.2.2 focused on memory ablation, here we benchmark against video diffusion baselines. As detailed in Tab. 4, baselines exhibit significantly higher errors as the horizon extends. Crucially, we interpret this degradation as a compound failure: since baselines frequently ignore control signals (as established in Sec. 4.2.1), their high LPIPS errors stem not only from spatiotemporal drift but also from insufficient action controllability, where the model fails to follow the inverse trajectory. In contrast, dWorldEval maintains high spatiotemporal consistency. Qualitative visualizations of these failure modes are provided in Appendix D.

## 5. Limitations

dWorldEval still has several limitations. Its progress predictor relies on predefined anchor states and VLM-distilled labels, which may not fully cover unconventional but valid solution paths, leading to imperfect intermediate progress estimates. In addition, rare unseen failures, particularly those involving fine contact timing, slippage, occlusion, or boundary-sensitive placements, remain challenging for an image-based world model without explicit contact dynamics. Our current analysis also focuses on success-rate ranking agreement rather than directly measuring whether policies induce identical action distributions under generated and real observations. Finally, the masked discrete diffusion formulation introduces nontrivial inference cost, motivating future work on token pruning, step distillation, speculative decoding, and more physically grounded modeling.

## 6. Conclusion

We presented dWorldEval, a discrete diffusion world model designed to overcome the limitations of existing methods in reliability. Our approach unifies action tokens with sparse keyframe memory to generate consistent long-horizon rollouts, quantified by our proposed $\Delta$LPIPS metric. Extensive experiments confirm that dWorldEval significantly enhances controllability, with predicted success rates that closely track real-world execution. This alignment enables accurate policy ranking across diverse architectures, bringing scalable evaluation closer to practice.

## Impact Statement

This paper presents work whose goal is to advance the field of Machine Learning. There are many potential societal consequences of our work, none which we feel must be specifically highlighted here.

## Acknowledgments

This work is supported by the Sci-Tech Innovation Initiative by the Science and Technology Commission of Shanghai Municipality (24ZR1419000), the National Science Foundation of China (12471501), and ECNU Multifunctional Platform for Innovation (001).

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

# A. More on Experimental Setup

In this section, we provide further details regarding the task definitions, data collection pipelines, and model hyperparameters used in our experiments.

## A.1. Detailed Task Descriptions

We evaluate dWorldEval across three domains: Real-World AgileX, RoboTwin (Mu et al., 2025), and LIBERO (Liu et al., 2023). Below we describe the specific task designs and their associated language instructions.

**Real-World AgileX Tasks** We collected data for five distinct tasks using a bimanual AgileX system. Initial states and corresponding instructions are visualized in Fig. 9.

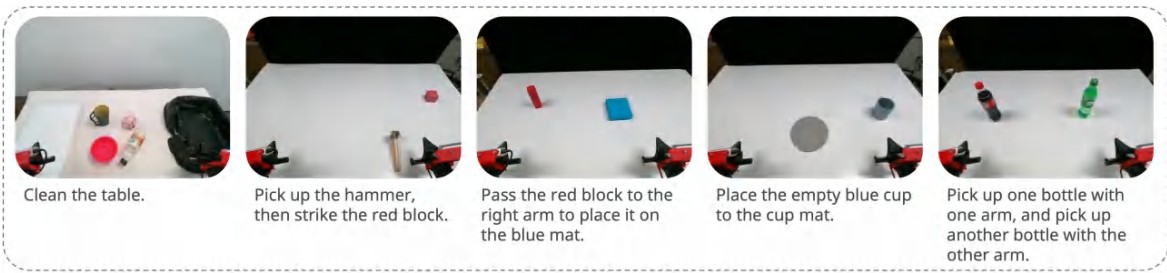

*Figure 9.* **Real-World Evaluation Tasks.** We visualize the initial scene observations and corresponding language instructions for the five bimanual manipulation tasks evaluated on the AgileX platform.

- **Bussing Table.** A long-horizon task requiring the robot to classify and sort multiple objects (tableware, trash) into trays or bins.
  *Instruction: "Clean the table."*

- **Place Cup.** Precision pick-and-place where the robot must align a cup onto a specific target mat.
  *Instruction: "Place the empty blue cup to the cup mat."*

- **Handover Block.** A bimanual coordination task involving passing a block from the left arm to the right arm before placement.
  *Instruction: "Pass the red block to the right arm to place it on the blue mat."*

- **Strike Block.** Dynamic tool manipulation that requires grasping a hammer to strike a target block.
  *Instruction: "Pick up the hammer, then strike the red block."*

- **Dual Bottle Pick.** A synchronization task requiring the robot to simultaneously grasp and lift two bottles positioned in front of it.
  *Instruction: "Pick up one bottle with one arm, and pick up another bottle with the other arm."*

**Simulation Tasks (RoboTwin & LIBERO)**

- **RoboTwin Stacking & Placement.** On the RoboTwin benchmark, we utilize the ARX dual-arm configuration to evaluate 10 diverse contact-rich tasks. We categorize these into: (1) **Precision Stacking**, which requires vertical stability and geometry alignment (e.g., Stack Blocks (Three), Stack Bowls (Two & Three)); and (2) **Constrained Placement**, which involves inserting objects into specific receptacles, such as placing Bread (into Basket/Skillet), Cans (into Basket/Pot/Plastic Box), Plate (into Container), and an Empty Cup. These scenarios involve complex inter-object occlusions and fine-grained physics that are challenging for video generation models.

- **LIBERO Suites.** We utilize four task suites from the LIBERO benchmark: LIBERO-Object, LIBERO-Spatial, LIBERO-Goal, and LIBERO-100. These tasks involve standard tabletop manipulation skills like opening drawers, moving objects around obstacles, and arranging items based on spatial relations.

*Table 5.* **Action-shuffle sanity check.** We permute future action chunks across samples while keeping history and language fixed. Shuffling breaks action–outcome alignment and degrades action-controllability indicators.

| Conditioning | LPIPS $\downarrow$ | $\Delta$LPIPS $\downarrow$ |
|---|---|---|
| Aligned actions | **0.215** | **0.324** |
| Shuffled actions | 0.461 | 0.697 |

## A.2. Model Implementation Details

dWorldEval is initialized from MMaDAVLA-8B, a bidirectional transformer with 32 layers, 32 attention heads, and a hidden dimension of 4096. The model is conditioned on a sparse history of $K = 4$ keyframes ($128 \times 128$). We use a prediction horizon $\Delta$ consistent with the action chunk size (randomly sampled from $[2, 8]$). The loss function balances visual reconstruction and progress prediction with weights $w_{\text{score}} = 2$ and $w_{\text{vis}} = 1$. The model was trained for 15 epochs on 8 H800 GPUs using the AdamW optimizer with a learning rate of 5e-5. We employ a global batch size of 128 (using gradient accumulation).

# B. Verifying Causal Dependency via Action Shuffling

A faithful action-conditioned world model should depend on the provided future action chunk. If we deliberately destroy the alignment between actions and outcomes, action-controllability indicators should drop accordingly.

**Experimental Protocol.** Given a test batch of samples $\{(o_t^i, h^i, l^i, a^i, o_{t+\Delta}^i)\}_{i=1}^{B}$, where $o_t^i$ is the current observation, $h^i$ is the observation history, $l^i$ the instruction, and $a^i$ the future action chunk, we construct a shuffled action input by permuting action chunks across the batch: $\tilde{a}^i \leftarrow a^{\pi(i)}$ with a random permutation $\pi$ (optionally enforcing $\pi(i) \neq i$). We then roll out the world model twice: (i) **Aligned**: $\hat{o}_{t+\Delta}^i \sim \mathcal{W}_\theta(o_t^i, h^i, l^i, a^i)$, (ii) **Shuffled**: $\tilde{o}_{t+\Delta}^i \sim \mathcal{W}_\theta(o_t^i, h^i, l^i, \tilde{a}^i)$. All evaluation metrics are computed identically under both settings. This perturbation preserves the visual/language marginal distribution while breaking the causal link between actions and future observations. In practice, we implement shuffling by a batch-wise permutation of action chunks, keeping all other inputs unchanged. We repeat the evaluation on the same test set used in Sec. 4.2.1. For clarity, we report the metrics averaged across the entire test set, as the degradation trend is consistent across both distributions.

**Results.** Tab. 5 shows that shuffling actions consistently degrades action-controllability indicators, confirming that our metric is sensitive to the action input and that the learned dynamics are not explained by static appearance priors alone.

# C. VLM Supervision Details

We utilize an off-the-shelf VLM (SEED-1.5VL (Guo et al., 2025a)) to generate ground-truth progress scores. Unlike standard zero-shot evaluation, we employ a few-shot In-Context Learning (ICL) strategy to align the model's scoring distribution with human intuition. For each query, we construct a prompt containing:

1. A detailed task definition and rigid scoring rules.

2. Three anchor examples with pre-labeled scores (e.g., 0.2, 0.4, and 0.6) to demonstrate intermediate states.

3. A batch of query frames (typically 10 frames) to be evaluated independently.

This batch-processing approach significantly stabilizes the output and enforces strict adherence to the discrete scoring criteria.

## C.1. Prompt Template

For the Libero-Object (Liu et al., 2023) suite, which primarily involves pick-and-place manipulation, we redefine the scoring criteria to reflect the sequential phases of the action. We employ an few-shot ICL strategy to help the VLM distinguish subtle state changes.

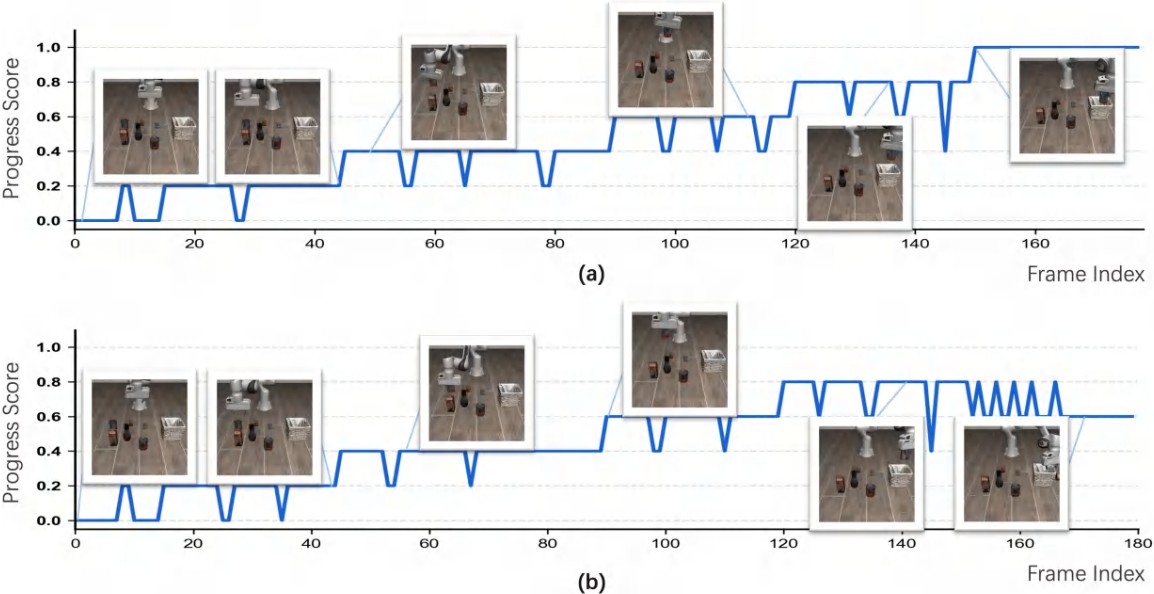

*Figure 10.* **Ground-truth Progress Labels.** Visualization of VLM-annotated scores for the LIBERO-Object (Liu et al., 2023) task *"pick up the alphabet soup and place it in the basket"*. (a) A successful trajectory exhibits step-wise score increases as milestones are achieved. (b) A failure trajectory effectively reflects incomplete execution, with the score stalling at a low value.

The system instruction maps the continuous manipulation process into discrete progress steps:

**System Instruction:**
You are an expert roboticist evaluator. Your task is to judge the completion progress of a robot performing a specific manipulation task (e.g., "pick up the bbq sauce and place it in the basket").
**Task Instruction:** {TASK_INSTRUCTION}
**Scoring Rules (Progress Phases):** The task progress is discretized into the following stages. Please choose the score that best matches the current visual state:

- **0.0**: *Idle / Start*. The robot has not yet interacted with the target object.
- **0.2**: *Approach / Contact*. The gripper is positioned near the target object or has just made contact, but the object is not yet lifted.
- **0.4**: *Lifted*. The object is successfully grasped and lifted off the surface.
- **0.6**: *In Transit*. The robot is moving the object towards the target area (mid-trajectory).
- **0.8**: *Pre-Placement*. The object is aligned with or hovering just above the target zone, ready for release.
- **1.0**: *Success*. The object is stably placed in the target configuration, and the gripper has released it (or the task is fully complete).

**Important note:** The input frames may appear in random order. You must evaluate each frame independently, strictly based on the visible state in that frame. Do not infer progress from previous frames.
**Valid outputs:** 0, 0.2, 0.4, 0.6, 0.8, 1.0
**Output Format:** Return only a list of numbers (e.g., [0.2, 0, 0.6, 1.0]). In-Context Examples:
We provide 8 distinct examples covering the full range of motion to anchor the scoring:
*[Image 1]* (Start State) → Text: "Example 1 score: 0.0"
*[Image 2]* (Approaching) → Text: "Example 2 score: 0.2"
*[Image 3]* (Just Lifted) → Text: "Example 3 score: 0.4"
*[Image 4]* (Mid-Air Moving) → Text: "Example 4 score: 0.6"
*[Image 5]* (Approaching Target) → Text: "Example 5 score: 0.6"
*[Image 6]* (Aligned/Hovering) → Text: "Example 6 score: 0.8"
*[Image 7]* (Just Released) → Text: "Example 7 score: 1.0"
*[Image 8]* (Retracting/Done) → Text: "Example 8 score: 1.0"
Text: "Now evaluate these frames:"

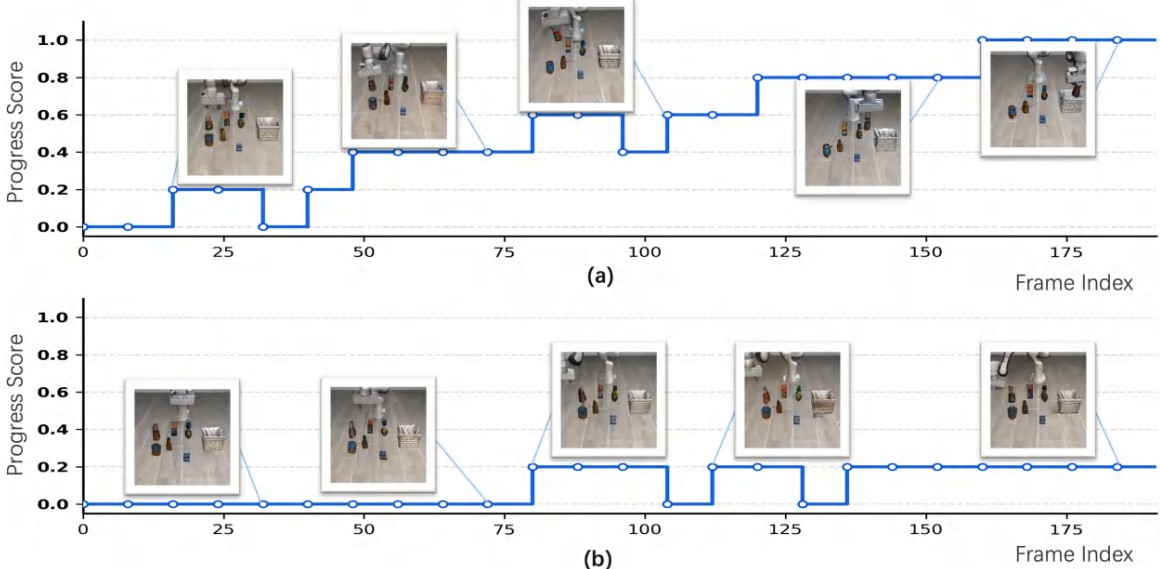

*Figure 11.* **Generated Progress Scores.** Visualization of scores predicted by dWorldEval for the LIBERO-Object (Liu et al., 2023) task *"pick up the ketchup and place it in the basket"*. (a) The model generates a successful rollout where the score accurately rises to 1.0. (b) The model faithfully predicts a failure case, maintaining a low score consistent with the visual outcome.

### C.2. Visualizing Progress Scores: Labels vs. Generation

To intuitively demonstrate the effectiveness of our Progress-as-text mechanism, we provide ground-truth annotations and model predictions. Figure 10 illustrates the ground-truth labeling process, where the VLM assigns discrete, step-wise scores (e.g., 0.2, 0.4) corresponding to specific achieved milestones. Complementing this, Figure 11 presents the generated progress scores from dWorldEval during inference. These results confirm that our model accurately learns to associate visual state transitions with the correct progress semantics.

## D. Visualizing Baseline Consistency

We provide the qualitative visualization corresponding to the round-trip analysis in Sec. 4.4. Figure 12 illustrates the generated rollouts on the LIBERO (Liu et al., 2023) suite with a horizon of $H = 20$. It can be observed that dWorldEval successfully restores the initial scene structure at the terminal step $t = 2H\Delta$. In contrast, the baselines (WorldEval and WorldGym) exhibit significant visual deviation from the initial state. This confirms that the high LPIPS errors reported in the main text stem from a compound failure: the baselines struggle not only with spatiotemporal drift (hallucinating objects) but also with strictly adhering to the inverse action sequence required to return to the start.

## E. More Visualization

In this section, we provide more visualizations for both the simulation and the real-world environment. Figure 13 presents the generated results on the RoboTwin (Mu et al., 2025) benchmark, and Figure 14 shows the real-world cases.

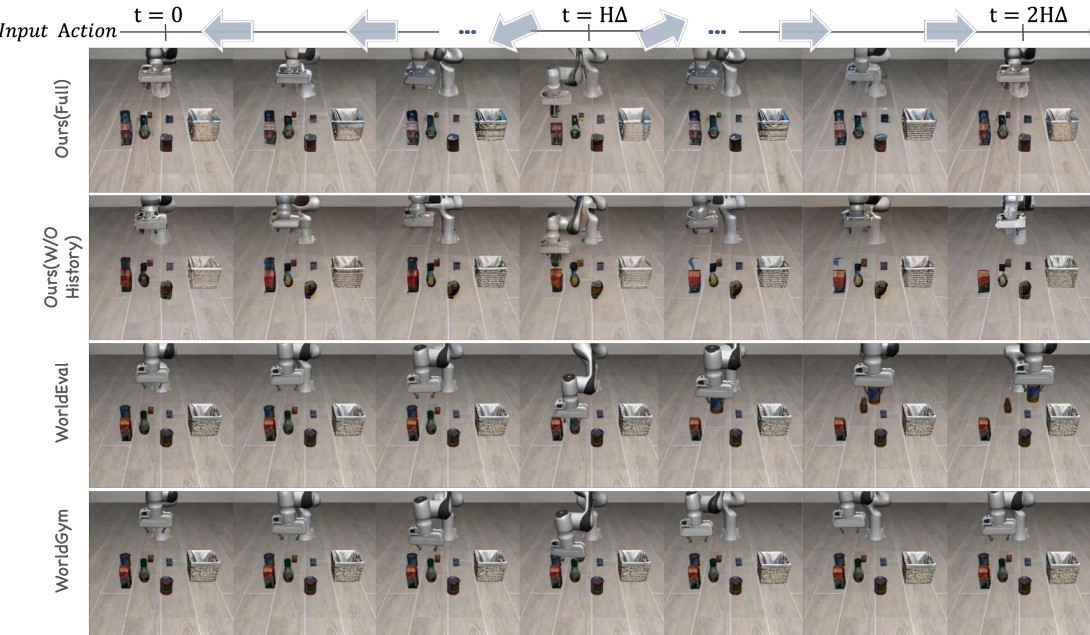

*Figure 12.* **Qualitative comparison of round-trip consistency with baselines.** We visualize the round-trip rollouts ($H = 20$) on the LIBERO (Liu et al., 2023) from the shared third-person view. We compare four models: WorldEval (Li et al., 2025b), WorldGym (Quevedo et al., 2025), dWorldEval (W/O History), and dWorldEval (Full). The goal is to return to the initial state at $t = 2H$ (rightmost column) by executing inverse actions. Ours (Full) successfully restores the initial scene structure. In contrast, baselines and the w/o-history ablation exhibit significant deviation at the end of the trajectory, caused by a combination of spatiotemporal drift and failure to follow the inverse control signals.

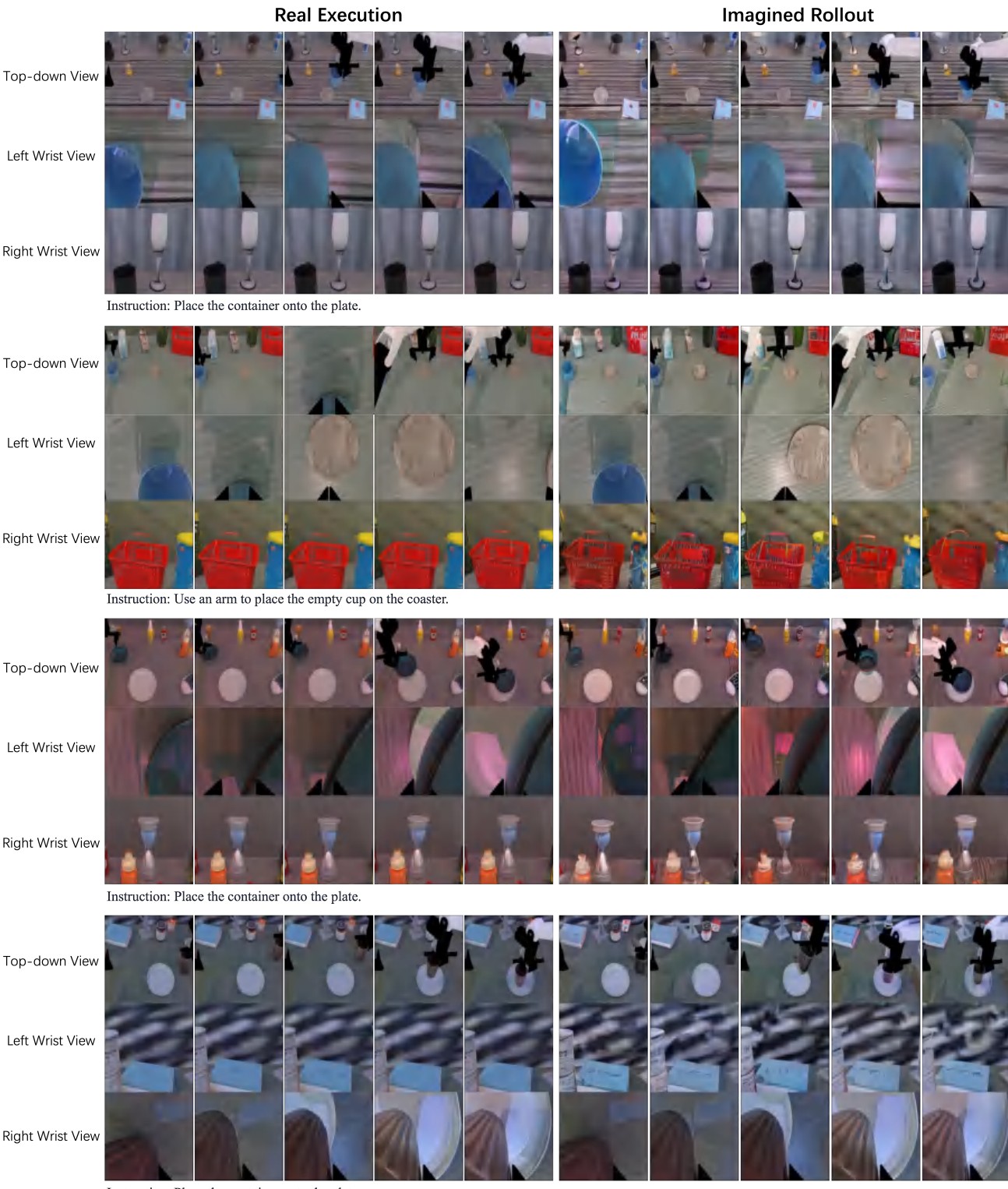

*Figure 13.* **Visualization on the RoboTwin benchmark.** ([Mu et al., 2025](#)) We compare the ground-truth simulation trajectory with the video generated by our model . Our model synthesizes high-fidelity, synchronized videos across three views (Top-down, Left Wrist, and Right Wrist), accurately preserving the object details and spatial layout of the simulation environment.

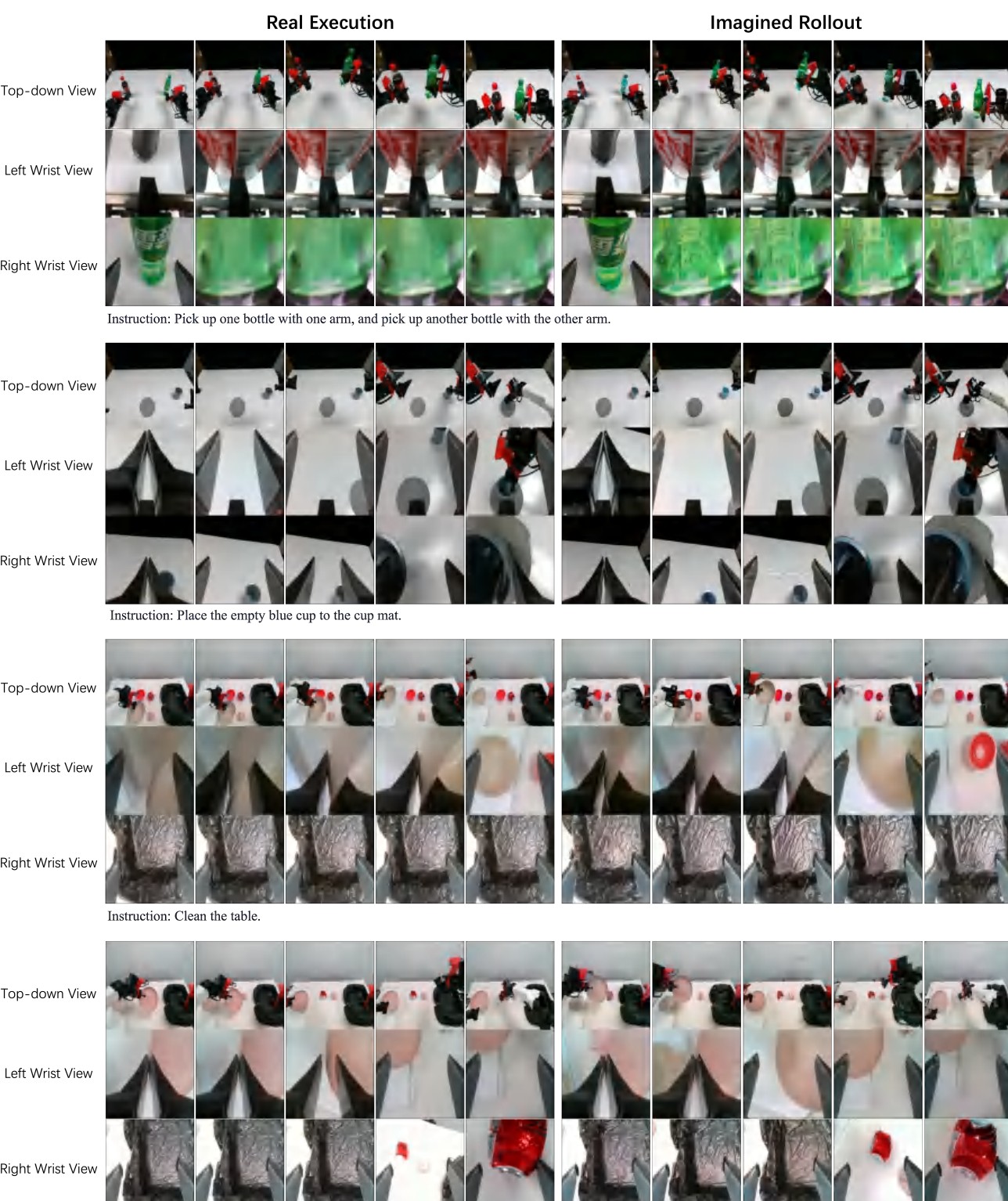

*Figure 14.* **Visualization in real-world scenarios.** We compare the ground-truth physical robot execution with the video generated by our model. Given the language instruction, our model synthesizes high-fidelity, synchronized videos across three views (Top-down, Left Wrist, and Right Wrist), accurately preserving object details and handling the visual complexity of the real-world environment.

