# OpenReview forum: "Scaling Real-World Robot Policy Evaluation via Discrete Diffusion World Model"
_ICML.cc/2026/Conference — ICML 2026 spotlight_

### Official Review · Reviewer_usH3 · 2026-03-04

**Soundness:** 3
**Presentation:** 3
**Significance:** 3
**Originality:** 2
**Overall Recommendation:** 5
**Confidence:** 4

**Summary:**

This paper studies scalable evaluation of generalist robot manipulation policies. The authors propose dWorldEval, an action centric discrete diffusion world model that tokenizes observations, language, and action chunks into a unified token space and denoises them with a single self attention backbone where actions are treated as first class tokens. To improve long horizon stability, the model incorporates a sparse keyframe memory. It further introduces Progress as text to jointly generate future observations and success indicators, enabling automatic policy scoring. Experiments on LIBERO, RoboTwin, and real robot tasks show improved action controllability, more stable rollouts, and high correlation between imagined and real world success rates.

**Compliance With Llm Reviewing Policy:**

Affirmed.

**Key Questions For Authors:**

The main text describes four real robot tasks, while the appendix lists five including Dual Bottle Pick. Which tasks are included in the real world Pearson correlation in Figure 7(d)? How many episodes per task are used? Please report per task real success rates with variance and, if possible, per task correlation values. If some tasks exhibit weaker correlation, please explain why.

How do Pearson correlation and ranking consistency change when using different VLMs for progress labeling, varying prompts or number of shots, or injecting controlled label noise? This analysis would clarify the robustness of Progress as text.

Are there cases where Delta LPIPS is low but the rollout is physically incorrect, for example in insertion or contact tasks? If so, how frequent are such failures and how do they affect ranking reliability?

**Limitations:**

No.

The paper would benefit from a clearer discussion of limitations, particularly regarding the dependence on VLM distilled progress supervision, potential teacher bias, and failure modes in contact rich tasks where perceptual metrics may not reflect physical correctness.

**Strengths And Weaknesses:**

Strengths

The paper addresses an important and practical problem in robotics, namely scalable policy evaluation without excessive real world testing. The action centric token unification is well motivated and directly targets a known weakness of prior video diffusion evaluators. The sparse keyframe memory is a reasonable design for reducing long horizon drift. Experiments cover multiple domains and include ablations and action perturbation tests that support the controllability claims. The use of correlation and ranking consistency as evaluation metrics is appropriate for the stated goal.

Weaknesses

First, the scope of the real world evaluation is unclear. The main text states four real robot tasks, while the appendix lists five including Dual Bottle Pick. It is not clear which tasks are included in the reported real world Pearson correlation and how many episodes per task are used. Per task statistics and variance are not provided. If some tasks show weaker correlation, this should be analyzed rather than reporting only an aggregate number.

Second, Progress as text relies on progress labels distilled from an external VLM. The paper does not analyze how sensitive Pearson correlation or ranking consistency is to different VLM choices, prompting strategies, or label noise. If the evaluator is highly teacher dependent, the claim of autonomy is weakened.

Third, Delta LPIPS measures perceptual change but may not reflect physical semantic correctness in contact rich tasks such as insertion or striking. The paper does not analyze whether low Delta LPIPS can still correspond to physically incorrect rollouts. This leaves uncertainty about whether the reliability bottleneck is fully resolved.

Overall, the method is technically solid and the empirical results are promising, but the above issues limit confidence in the strongest claims.

---

> ### Author Rebuttal · Authors · 2026-03-31
>
> Dear Reviewer usH3,
>
> We sincerely appreciate your thoughtful comments and constructive critique. Our detailed point-by-point responses to your questions are provided below.
>
> 1. ### Real-World Evaluation Details
>
> We sincerely thank the reviewer for pointing out this inconsistency, and we apologize for the confusion caused by our wording. The “four-task” description in Sec. 4.1 is a typo; Fig. 7(d) was in fact evaluated on all five real-world tasks, including Dual Bottle Pick, with 30 episodes per policy for each task .Below, we report the Pearson $r$ and the success rates (Real vs. WM, $\pm$ SE).across policies (π0, DexVLA, and DP):
>
> | **Task (Pearson r)**         | **pi0 (Real / WM)** | **DexVLA (Real / WM)** | **DP (Real / WM)**  |
> | ---------------------------- | ------------------- | ---------------------- | ------------------- |
> | **Bussing table** (0.858)    | 0.85±.06 / 0.70±.07 | 0.70±.07 / 0.85±.06    | 0.35±.08 / 0.25±.07 |
> | **Dual Bottle Pick** (0.817) | 0.40±.08 / 0.25±.07 | 0.10±.05 / 0.20±.06    | 0.00±.00 / 0.00±.00 |
> | **Place empty cup** (0.987)  | 0.55±.08 / 0.60±.08 | 0.35±.08 / 0.45±.08    | 0.10±.05 / 0.15±.06 |
> | **Strike block** (0.971)     | 0.55±.08 / 0.45±.08 | 0.55±.08 / 0.55±.08    | 0.05±.03 / 0.15±.06 |
> | **Handover block** (0.786)   | 0.25±.07 / 0.30±.07 | 0.20±.06 / 0.10±.05    | 0.00±.00 / 0.05±.03 |
>
> As shown in the table, the Handover block task exhibits a relatively lower correlation (r = 0.786). We believe this is expected given the nature of the task: it requires precise bimanual transfer, stable cross-arm contact, and a boundary-sensitive terminal placement, making it substantially more sensitive than the other tasks to small errors in contact timing and object pose. Two failure modes are particularly important. First, false positives can arise because an image-based world model without explicit contact dynamics may fail to faithfully reproduce abrupt slips or unstable mid-air transfer failures. Second, false negatives can occur in near-boundary placements: when the real execution barely succeeds, a slight spatial deviation in the generated terminal frame may visually place the object just outside the target region, flipping the success label. We will clarify this point in the revision and explicitly note that such contact-critical bimanual tasks are the most challenging regime for world-model-based evaluation.
>
> ### 2. Robustness of Labeling
>
> We agree this should be analyzed more carefully. In the rebuttal, we tested sensitivity to both the teacher VLM and the number of in-context shots. Using SEED-1.5VL, performance degrades gradually from 8-shot (r=0.910) to 5-shot (r=0.892) and 1-shot (r=0.706), indicating that the method does not depend on a single rigid prompt template. Replacing SEED-1.5VL with Qwen2.5-VL-7B yields a modest drop (r=0.865), while preserving the overall policy ranking trend. We will include these results in the revision.
>
> ### 3. Low $\Delta$-LPIPS with Physically Incorrect Rollouts
>
> Such cases do exist, but they are infrequent and mostly arise in contact-sensitive or partially occluded settings. One example is Strike Block, where the hammer can visually occlude a near-miss contact, leading to a low  $\Delta$-LPIPS despite physical failure. We have observed a small number of such edge cases, but did not systematically quantify their frequency in the current paper. Importantly, $\Delta$-LPIPS is used as a diagnostic of action-conditioned dynamic fidelity rather than the sole success criterion, and the final policy evaluation is based on multi-view rollout assessment of the generated outcomes rather than this metric alone. In practice, the use of three synchronized views helps reduce such ambiguities by cross-checking fine-grained contacts from complementary viewpoints, so these cases mainly introduce local noise rather than a systematic ranking bias. This is also consistent with the strong ranking correlation we observe in the multi-view and real-world settings.
>
> We hope this comprehensive data and analysis address your concerns. Please let us know if there is anything else you would like us to clarify!

---

> > ### Author Rebuttal · Reviewer_usH3 · 2026-04-02
> >
> > I thank the authors for the detailed per-task statistics, VLM robustness analysis, and clarification on Delta-LPIPS limitations. My concerns have been adequately addressed. I maintain my current score.

---

### Official Review · Reviewer_Lo5o · 2026-03-09

**Soundness:** 3
**Presentation:** 3
**Significance:** 3
**Originality:** 3
**Overall Recommendation:** 4
**Confidence:** 4

**Summary:**

In this paper, the authors introduce a world model-based framework for scalable robot policy evaluation, avoiding costly real-world rollouts. Prior video diffusion world models treat actions as weak auxiliary conditions via cross-attention or AdaLN, causing strong visual priors to override control signals and hallucinate successful outcomes. dWorldEval instead tokenizes visual observations, language instructions, and action chunks into a unified discrete token space and denoises them jointly through a single bidirectional transformer, making actions first-class citizens in the generation process. In addition, the authors use a sparse keyframe memory to anchor long-horizon spatiotemporal consistency, and a progress-as-text mechanism that jointly generates future observations and task scores. Experiments are conducted on LIBERO, RoboTwin, and a real bimanual AgileX robot, showing strong pearson correlation (r ≈ 0.9) between estimated and real success rates.

Overall, the paper tackles an interesting and practically important problem of policy evaluation at scale. The paper is well-written and the core diagnosis of the action-conditioning failure mode is well-supported. Experiment results, especially the real-world results across diverse tasks, are impressive.

**Compliance With Llm Reviewing Policy:**

Affirmed.

**Final Justification:**

The authors have effectively addressed my concerns during the rebuttal. My overall feedback of the paper remains positive.

**Key Questions For Authors:**

* Can the authors provide more details on the model's architecture, especially the exact representation of discrete tokens, and why discretization helps with dynamics modeling?
* What is the inter-rater agreement between SEED-1.5VL progress scores and human annotators on the same frames? Have you tested with a different VLM grader to check robustness of the distilled labels?
* The pearson correlation test measures rank-order agreement between estimated and real success rates, but it does not capture whether the policy behaves the same under generated vs. real observations. Have you measured whether the policy produces qualitatively similar action distributions when fed generated frames vs. real frames?

**Limitations:**

The authors should include a more comprehensive limitation section. The authors are also encouaged to write about potential future directions for the method.

**Strengths And Weaknesses:**

Strengths
* The central claim that prior world models treat actions as weak conditions, allowing visual priors to dominate and hallucinate successes, is clearly stated and shown in the evaluation. The proposed fix of making actions first-class tokens directly addresses this architectural root cause rather than patching it with more data.
* Strong empirical results across diverse settings. The correlation between estimated and real success rates is impressive across different policy architectures and across domains. The real-world bimanual AgileX experiments across five tasks is a highlight.
* Clear and well-structured paper. The motivation, architecture, and evaluation protocol are logically organized, and the round-trip consistency probe is an elegant diagnostic design.

Weaknesses
* The causal attention structure is underspecified and raises some concerns. The transformer uses bidirectional attention, meaning action tokens at future timesteps are also visible when generating intermediate frames. It is unclear whether the model can attend to future action tokens when generating earlier frames, which would break the causal assumption.
* Progress-as-text supervision relies on a proprietary VLM. The training labels for progress scores are generated by SEED-1.5VL, a closed-source model, making the model dependent on another VLM evaluator.
* Inference speed is a significant bottleneck. The paper reports 1.5 seconds per frame and 30 to 90 seconds for a full trajectory evaluation. For evaluation to be a practical alternative to real-world rollouts, the world model needs to be at least comparable in wall-clock cost. The paper does not discuss this limitation or explore ways to accelerate inference.
* The long-horizon consistency test assumes the environment is fully reversible, which is not true for tasks involving contact or object placement. The protocol may overestimate consistency for tasks where the physical reverse action does not produce the same visual outcome as the original initial frame.

---

> ### Author Rebuttal · Authors · 2026-03-31
>
> Dear Reviewer Lo5o,
>
> Thank you for your careful review and valuable suggestions. We address your questions below.
>
> ### 1. Clarification on Causality and the Role of Discrete Tokenization
>
> Thank you for pointing this out. dWorldEval is not a per-frame autoregressive video predictor. At each rollout step, it predicts a single horizon-$\Delta$ future observation (and progress score) conditioned on the current observation, history, instruction, and current action chunk. Therefore, the concern that the model may use “future” action tokens to generate earlier intermediate frames does not apply here.
>
> Bidirectional attention is confined to this one-step prediction and does not expose the model to future rollout actions. After predicting $\hat{o}_{t+\Delta}$ the policy receives this new observation and only then produces the next action chunk. Thus, there is no leakage from future rollout actions into the current prediction. We will clarify in the revision that our “causal” claim refers to action-conditioned closed-loop rollout causality, rather than frame-by-frame temporal masking within a single prediction step.
>
> ### 2. Inter-rater Agreement for Progress Scores and VLM Robustness
>
> Thank you for this important question. We evaluated label reliability on 40 LIBERO and 40 real-world rollout episodes by asking human annotators and SEED-1.5VL to score the same extracted frames independently. Since the labels are ordinal, we report Linear Weighted Kappa (LWK) and Pearson correlation, obtaining high agreement on LIBERO (0.85 / 0.88) and real-world data (0.82 / 0.86), respectively.
>
> We also replaced SEED-1.5VL with Qwen2.5-VL-7B to test teacher-model robustness. The resulting evaluation correlation decreases from (r=0.910) to (r=0.865), while preserving the overall policy ranking. We will add these results in the revision.
>
> ### 3. Policy Action Distribution on Generated vs. Real Frames
>
> Thank you for this insightful question. We agree that Pearson correlation measures success-rate ranking agreement, but does not directly establish whether the policy induces the same action distribution under generated versus real observations. We did not explicitly evaluate action-distribution similarity in the current paper, and agree this would be a valuable additional diagnostic.
>
> Our current evidence is therefore indirect but relevant to the closed-loop setting we study. Since the policy repeatedly acts on model-generated observations, reliable ranking requires those observations to remain action-faithful and temporally consistent. Consistent with this, Table 1 / Fig. 5(a) show that dWorldEval better preserves action-conditioned outcomes, especially on failure cases, while Table 2 / Fig. 5(b) show substantially less long-horizon drift. Under these conditions, estimated success rates remain highly correlated with real execution across LIBERO multi-view, RoboTwin, and real-world tasks in Fig. 7(b–d). Moreover, when action–outcome alignment is deliberately corrupted, both controllability and ranking correlation degrade sharply (Fig. 8 / Table 5). While this does not establish exact action-distribution equivalence, it supports the more limited claim that dWorldEval preserves policy ordering over the evaluated horizons. We will clarify this limitation in the revision.
>
> ### 4. Inference Speed Bottleneck for Large-scale Evaluation
>
> Thank you for this important point. We agree that inference latency is a real limitation of the current system, and we will make this explicit in the revised paper. In the current manuscript, we report approximately 1.5 s/frame and 30–90 s for a full trajectory evaluation on a single H800 GPU.
>
> That said, our goal in this paper is not to claim that the current implementation is already optimal in absolute latency, but to show that a world model can serve as a **reliable,** **automated policy-in-the-loop evaluator** that is substantially easier to scale than repeated real-world rollouts. In practice, real-world evaluation also incurs robot reset time, human monitoring, and safety overhead, while world-model evaluation can be run unattended and parallelized across GPUs.
>
> We also agree that accelerating inference is an important future direction. Given our long-sequence unified formulation and masked discrete diffusion architecture, promising directions include token pruning, step distillation, and speculative decoding. We will add this limitation and these acceleration opportunities to the discussion section.
>
> ### 5. Limitations of the Round-Trip Evaluation Metric
>
> Thank you for this sharp observation. We agree that contact-rich physical interactions are rarely reversible, so the round-trip metric may overestimate absolute consistency. We therefore use it only as a relative diagnostic for comparing long-horizon drift and action-following behavior across models, and will clarify this limitation in the revision.
>
> We hope these clarifications address your concerns.

---

> > ### Author Rebuttal · Reviewer_Lo5o · 2026-04-02
> >
> > I thank the authors for their detailed explanations, including the clarification on causality of their world model, additional results on inter-rate aggrements, and limitation discussions. I would like to keep my score. My overall feedback of the paper remains positive.

---

### Official Review · Reviewer_vwoP · 2026-03-11

**Soundness:** 3
**Presentation:** 4
**Significance:** 4
**Originality:** 3
**Overall Recommendation:** 5
**Confidence:** 4

**Summary:**

The paper introduces dWorldEval, an action-centric discrete diffusion world model designed to automate the evaluation of robotic manipulation policies. To address the issue of continuous video diffusion models hallucinating task successes or ignoring control signals, the authors propose mapping visual observations, language instructions, and action chunks into a unified discrete token space. This allows the model to process actions as causal drivers through a single self-attention backbone using Masked Discrete Diffusion (MDD). Furthermore, the model incorporates a sparse keyframe memory to anchor global spatiotemporal consistency across long rollouts , and introduces a "Progress-as-text" mechanism. This mechanism allows the model to jointly predict future visual states alongside a discrete success score, removing the need for external evaluators at inference.

**Compliance With Llm Reviewing Policy:**

Affirmed.

**Key Questions For Authors:**

To strengthen the paper's claims, I have a few key questions regarding the methodology and evaluation. First, regarding the "Progress-as-text" module, how does dWorldEval score an imagined rollout if a policy successfully achieves the final goal state but does so via an erratic or unconventional trajectory that bypasses the human-defined intermediate anchor states (e.g., "0.4: Lifted," "0.6: In Transit") hardcoded into the teacher VLM's prompt? Second, since the model learns failure dynamics from 1k human-collected failures, have you observed instances during real-world evaluation where dWorldEval fails to accurately simulate a policy-induced failure because the AI's specific kinematic error was unrepresented in the human-teleoperated dataset? Third, addressing scalability, given the 30-90 second generation time per trajectory using 16-step decoding on an H800 GPU, what are your plans to incorporate techniques like latent consistency distillation or token-pruning to improve throughput for large-scale evaluation? Finally, while the action-shuffling ablation proves the proposed A-LPIPS metric is sensitive to action inputs, do you have quantitative data demonstrating that it strongly correlates with human perceptual judgments of dynamic transition quality?

**Limitations:**

The authors adequately mention the lack of specific societal consequences, but they do not sufficiently address the methodological limitations of their VLM distillation pipeline. The reliance on human-curated anchor examples and strict, hardcoded text prompts to teach the model "progress" introduces a significant human expert bias that should be explicitly discussed as a limitation in the text. Furthermore, the high computational cost of inference (up to 90 seconds per rollout on premium hardware) should be contextualized regarding the claim of "scalability."

**Strengths And Weaknesses:**

dWorldEval's main strength is its unified discrete token space, which processes vision, language, and actions together to ensure the model strictly follows control signals and avoids hallucinating successes. Furthermore, its innovative "Progress-as-text" feature allows the model to intrinsically score task success by co-generating visual futures and discrete progress tokens, eliminating the need for external evaluators at inference. It also successfully mitigates long-horizon spatiotemporal drift using a multi-resolution sparse keyframe memory.

However, the methodology introduces significant human bias during training, as the "Progress-as-text" labels are generated by a VLM constrained by rigid, human-defined progress phases and curated anchor images. This risks penalizing valid AI policies that employ unconventional kinematics. Additionally, claims of evaluating "generalist" policies are overstated, as the real-world validation covers only four tasks and relies on human-teleoperated failures that may not accurately reflect autonomous AI errors. Finally, true scalability is limited by high computational costs, with inference taking 30 to 90 seconds per trajectory on a premium H800 GPU.

---

> ### Author Rebuttal · Authors · 2026-03-31
>
> Dear Reviewer vwoP,
>
> We sincerely thank you for your insightful review and highly constructive feedback. Following your suggestion, we will explicitly add a dedicated Limitations section in the revised manuscript to discuss the human bias introduced by VLM distillation and the computational cost of inference. Below, we address your detailed questions point by point.
>
> ### 1. Scoring on Unconventional Trajectories
>
> If a policy executes an unconventional trajectory that bypasses our predefined anchor states, the intermediate progress scores will likely exhibit stagnation or fluctuation. However, because our automated success criterion relies strictly on the predicted score of the terminal frame reaching 1.0, these intermediate deviations do not compromise the final binary evaluation of task success.
>
> We fully acknowledge this limitation. For open-ended tasks with multiple solutions, manually defined anchors inevitably fail to capture all valid transitional paths. We will explicitly add a discussion on this limitation in the revised manuscript.
>
> ### 2. Generalizing to Unseen Policy Failures
>
> Thank you for this important question. We agree that this is a critical test of whether dWorldEval captures policy-induced failure dynamics, rather than merely memorizing the 1k human-collected failure trajectories. We would like to clarify that these failure data are included to expose the model and the progress predictor to non-successful outcomes, but they are not intended to exhaust the space of possible real-world failure modes.
> More importantly, our evidence is not limited to those human failures. In the supplementary video (13s–28s), we show LIBERO examples where the policy exhibits erroneous action patterns that are not represented in the human-collected failure set, yet dWorldEval still faithfully reproduces the resulting failed behaviors instead of self-correcting them into successful outcomes. This behavior is consistent with the design of dWorldEval: it is trained natively as an action-centric world model that jointly models observations, language, and actions in a unified token space. As a result, its predictions are driven more directly by the input action sequence, rather than by matching previously seen failure appearances. Therefore, while rare fine-grained unseen kinematic errors may still remain challenging—especially over long horizons—the current results suggest that dWorldEval is not simply overfitting the human-collected failure cases, but is learning a more general action-faithful failure model.
>
> ### 3. Optimizing Inference Speed
>
> Thank you for this important point. We agree that inference latency is a real limitation of the current system, and we will make this explicit in the revised paper. In the current manuscript, we report approximately 1.5 s/frame and 30–90 s for a full trajectory evaluation on a single H800 GPU.
> That said, our goal in this paper is not to claim that the current implementation is already optimal in absolute latency, but to show that a world model can serve as a reliable, automated policy-in-the-loop evaluator that is substantially easier to scale than repeated real-world rollouts. In practice, real-world evaluation also incurs robot reset time, human monitoring, and safety overhead, while world-model evaluation can be run unattended and parallelized across GPUs.
> We also agree that accelerating inference is an important future direction. Given our long-sequence unified formulation and masked discrete diffusion architecture, promising directions include token pruning, step distillation, and speculative decoding. We will add this limitation and these acceleration opportunities to the discussion section.
>
> ### 4. Correlation of  $\Delta$-LPIPS with Human Perceptual Judgments
>
> We thank the reviewer for this excellent suggestion. To quantitatively verify the alignment between our proposed $\Delta$-LPIPS  metric and human perception, we conducted an additional evaluation. Setup: Human evaluators provided binary judgments ("Acceptable" vs. "Contains Artifacts") on the transition quality of 200 generated samples. Results: Evaluating the continuous  $\Delta$-LPIPS scores against these ground-truth human judgments yielded a high **AUROC of 0.89**. This quantitatively confirms that  $\Delta$-LPIPS serves as a highly accurate proxy for human perceptual judgments of dynamic transition quality.
>
> We hope this response addresses your concerns. Please let us know if there are any further questions during the discussion period!

---

> > ### Author Rebuttal · Reviewer_vwoP · 2026-03-31
> >
> > I acknowledge that the authors have addressed the majority of my concerns substantively. The A-LPIPS human correlation data (AUROC 0.89) is convincing, and I appreciate the honest concession regarding unconventional trajectory scoring and the commitment to an explicit Limitations section.
> > However, I have two follow-up points before I can consider my concerns fully resolved:
> > First, regarding Progress-as-text robustness: the authors' response to Reviewer usH3 reveals that 1-shot labeling drops Pearson correlation to r=0.706, a substantial degradation. This is directly relevant to the human bias concern I raised. While 8-shot performance is strong, the sensitivity to in-context shot count suggests the distillation pipeline's reliability is tightly coupled to prompt engineering quality. Could the authors clarify whether the anchor images used across all experiments were selected via a systematic protocol or curated per-task by an expert? If the latter, this dependency should be foregrounded in the Limitations section alongside the VLM choice sensitivity, as it represents an additional axis of human expert bias in the evaluation pipeline.
> > Second, regarding unseen failure generalization: I accept the architectural argument that first-class action tokens should improve generalization beyond memorized failures. However, the supplementary video evidence is anecdotal. Given that the authors have per-task real-world data available, could they identify even one concrete instance where a policy-induced failure mode was both (a) absent from the 1k human-collected set and (b) correctly predicted by dWorldEval, with enough specificity to verify the claim? A single well-documented case would be more persuasive than the general architectural argument.
> > These are clarification requests rather than fundamental objections. My overall assessment of the paper remains positive.

---

> > > ### Author Response · Authors · 2026-04-04
> > >
> > > We thank the reviewer for the constructive follow-up and appreciate the opportunity to clarify these two remaining points.
> > > First, the anchors are task-specific, because the progress phases themselves are task-specific. However, they were not curated post hoc for each experiment. For each task, we first defined canonical stages, assembled candidate anchors/examples for those stages, ran a small pre-experimental calibration with the teacher VLM solely to ensure consistency with the predefined rubric. That is all. This set was frozen before any downstream policy-ranking experiments and reused for all policies, checkpoints, and rollouts of that task, without per-experiment tuning. At the same time, we agree with the reviewer’s broader concern: the 1-shot degradation is important, because it shows that Progress-as-text depends not only on the teacher VLM, but is also sensitive to the design of the task-specific anchors/examples and scoring prompt. We will therefore make this dependency explicit in the limitations section, as an additional axis of human expert bias in the evaluation pipeline.
> > > We provide two concrete real-world cases in the [anonymous video](https://anonymous.4open.science/r/demo-3CDC/0401.mp4): Place Cup (00:00–00:08) and Strike Block (00:09–00:18). Our human-failure collection already included broad outcomes such as missed grasps/failed contacts and incorrect placement, so our claim is not that these categories were missing from the dataset. Rather, in a manual review of the task-specific collected failures, we did not observe the two behaviors shown here: in Place Cup, the policy reaches the cup mat and then makes several short retries near the target without settling the cup; in Strike Block, the policy jitters in place before a series of failed hammer strikes. Starting from the same initial frame and using the same executed action sequence, dWorldEval reproduces both behaviors as well as the same failed outcome. We do not claim that this proves coverage of all unseen failures, but these cases do show that dWorldEval is not limited to replaying the exact failure cases seen during collection.
> > > We hope this clarification addresses the reviewer’s remaining concerns.

---

### Official Review · Reviewer_ErW8 · 2026-03-11

**Soundness:** 4
**Presentation:** 4
**Significance:** 3
**Originality:** 3
**Overall Recommendation:** 4
**Confidence:** 4

**Summary:**

In this paper, the authors propose dWorldEval, a robot policy evaluation framework driven by an action-centric discrete diffusion world model. To effectively capture global scene context and task progress, the authors introduce a sparse keyframe memory module. Additionally, they propose a "Progress-as-text" mechanism to simultaneously generate future observations and success signals. Empirical results demonstrate a positive correlation between the evaluation metrics derived from dWorldEval and the actual success rates in real-world executions.

**Compliance With Llm Reviewing Policy:**

Affirmed.

**Final Justification:**

The author’s clarifications during the rebuttal phase have further strengthened the paper’s persuasiveness, and I am inclined to accept it. However, due to the lack of benchmark results testing generalizability, a higher score is unlikely.

**Key Questions For Authors:**

1. While the results on LIBERO are positive, it is generally acknowledged that LIBERO has limited visual and trajectory diversity, which can make models evaluated on it prone to overfitting. To better demonstrate the robustness and zero-shot generalization capabilities of the proposed world model, I strongly suggest that the authors evaluate dWorldEval on more diverse benchmarks, such as LIBERO-Plus (or similar environments).

2. To firmly establish the necessity and superiority of utilizing a generative world model for policy evaluation, it is crucial to compare dWorldEval with "model-free" progress-based or value-based evaluation methods (e.g., SARM, GVL, VIP). Including these baselines would help clarify whether the heavy machinery of a discrete diffusion world model provides a significant empirical advantage over standard pre-trained visual representations for evaluating task progress and policy performance.

3. Minor Issue: In line 250, the tabletop manipulation tasks in RoboTwin are described as "contact-rich." However, the tasks in this benchmark typically do not fall under the strict definition of contact-rich manipulation (which generally implies tasks involving complex force dynamics, such as tight-tolerance peg insertion or assembly). I suggest revising this statement to ensure technical accuracy.

**Limitations:**

yes

**Strengths And Weaknesses:**

# Strengths:
The paper is well-structured, clearly written, and the methodology is logically presented. The experimental results on the LIBERO benchmark are promising and demonstrate the feasibility of the proposed evaluation framework.

# Weaknesses:
While the proposed framework is interesting, my primary concerns pertain to the comprehensiveness of the empirical evaluation. Specifically, I have reservations regarding the complexity of the chosen benchmark and the absence of comparisons with model-free policy evaluation baselines. Please refer to the detailed questions below.

---

> ### Author Rebuttal · Authors · 2026-03-31
>
> Dear Reviewer ErW8,
>
> We sincerely thank you for your time, constructive feedback, and valuable suggestions. Below, we address your comments in detail.
>
> ### 1. Generalization Beyond LIBERO
>
> Thank you for this helpful suggestion. We agree that evaluating only on LIBERO would be insufficient to support strong claims about robustness and generalization, given its limited visual and trajectory diversity. Our intent, however, was precisely to go beyond a single benchmark. In addition to the four LIBERO suites, we evaluate dWorldEval on RoboTwin, which contains diverse bimanual manipulation tasks, and on a physical real-world AgileX platform with synchronized multi-view observations. Across these substantially different domains, dWorldEval still maintains strong agreement with real execution, achieving correlations of 0.910 on LIBERO multi-view, 0.927 on RoboTwin, and 0.918 on real-world tasks. We believe this already provides encouraging evidence that the proposed evaluator is not merely fitting LIBERO-specific visual or trajectory statistics, but can remain predictive under substantial shifts in embodiment, viewpoint, and task composition. While LIBERO-Plus would be a valuable additional benchmark, we are unfortunately unable to include a full evaluation on it in the revision given the limited rebuttal window. We will clarify this limitation in the revised manuscript and leave LIBERO-Plus evaluation to future work.
>
> ### 2. Comparison with Model-Free Evaluation Methods
>
> We thank the reviewer for pointing out VIP, GVL, and SARM as relevant prior work. We agree that these methods are important baselines for post-hoc rollout scoring, since they estimate reward/progress/value from an already observed trajectory or video. However, they are not policy-in-the-loop evaluators in the setting considered in our paper.
>
> Our method, like the Ctrl-World and World-Gym approaches, evaluates a policy by rolling it out in a learned action-conditioned world model: starting from the current observation, the policy produces an action, the world model generates the next observation, and this closed-loop process continues online. In contrast, VIP/GVL/SARM do not themselves generate future observations conditioned on policy actions; instead, they score an already available rollout. Therefore, directly comparing them to our method as if they solved the same task would conflate two distinct evaluation protocols: (1) post-hoc scoring of observed rollouts and (2) closed-loop policy evaluation through interactive world simulation.
>
> In particular, using VIP/GVL/SARM for policy evaluation requires first collecting rollouts from the target policy in the real world or an external simulator, whereas our setting is precisely motivated by reducing this dependence and evaluating policies directly through imagined interaction in the learned world. We do not argue that VIP/GVL/SARM are irrelevant; rather, they are baselines for a different point in the evaluation pipeline.
>
> ### 3. Terminology Correction
>
> We sincerely thank you for your careful reading and for pointing out this terminology issue. You are absolutely right; the tabletop tasks in RoboTwin (such as stacking and placement) do not involve the complex force dynamics or tight tolerances typically implied by the strict definition of "contact-rich" manipulation. We gladly accept your suggestion to ensure technical accuracy. In the revised manuscript, we will update Line 250 (and any other relevant instances) to describe these tasks as "multi-object tableware manipulation" rather than "contact-rich." We appreciate your attention to detail, which significantly helps improve the precision of our writing.
>
> We hope these clarifications thoroughly address your concerns, and we are happy to answer any further questions during the discussion period.

---

> > ### Author Rebuttal · Reviewer_ErW8 · 2026-04-03
> >
> > Thanks for your response. My concerns have been addressed. I will keep my score to recommend acceptance.

---

### Decision · Program_Chairs · 2026-04-30

**Decision:**

Accept (spotlight)

**Comment:**

The paper introduces dWorldEval, an action-centric discrete diffusion world model designed to automate the evaluation of robotic manipulation policies without requiring costly real-world rollouts. The framework is evaluated across LIBERO, RoboTwin, and real-world bimanual tasks, demonstrating a strong correlation between the model's automated success estimates and actual real-world execution rates.

Reviewers universally agreed that the paper addresses a critical and practical problem in robot learning by directly targeting the architectural flaws of prior video diffusion evaluators. The empirical results were highlighted as a major strength, particularly the strong correlation metrics achieved on real-world, bimanual hardware. However, reviewers raised valid concerns regarding the computational bottleneck of inference, the reliance on a proprietary teacher VLM for progress labeling, and ambiguities in the real-world task breakdown. During the rebuttal, the authors thoroughly resolved these issues by providing an explicit breakdown of all five real-world tasks, running additional human-correlation studies for their A-LPIPS metric, and testing open-source VLM alternatives to demonstrate label robustness.

Overall, the authors propose a highly effective and well-motivated architectural shift for world-model-based policy evaluation. I recommend acceptance.